# The Synthesis and Optical Property of a Ternary Hybrid Composed of Aggregation-Induced Luminescent Polyfluorene, Polydimethylsiloxane, and Silica

**DOI:** 10.3390/polym16233331

**Published:** 2024-11-27

**Authors:** Nurul Amira Shazwani Zainuddin, Yusuke Suizu, Takahiro Uno, Masataka Kubo

**Affiliations:** Division of Applied Chemistry, Graduate School of Engineering, Mie University, 1577 Kurimamachiya-cho, Tsu 514-8507, Japan; 422de02@m.mie-u.ac.jp (N.A.S.Z.); 423m327@m.mie-u.ac.jp (Y.S.); uno@chem.mie-u.ac.jp (T.U.)

**Keywords:** aggregation-induced emission, polyfluorene, organic/inorganic hybrid, sol-gel method

## Abstract

Tetraphenylethene (TPE) is known as a molecule that exhibits aggregation-induced emission (AIE). In this study, pendant hydroxyl groups were introduced onto polyfluorene with a TPE moiety. Sol-gel reactions of polydiethoxysiloxane (PDEOS) were carried out in the presence of hydroxyl-functionalized AIE polyfluorene (TPE-PF-OH) and polydimethylsiloxane carrying pendant hydroxyl groups (PDMS-OH) to immobilize AIE polyfluorene into a PDMS/SiO_2_ hybrid in an isolated dispersion state. The luminescence intensity from this three-component hybrid increased with the increase in silica content. The luminescence intensity decreased with increasing external temperature. For the control experiment, sol-gel reactions of PDEOS were carried out in the presence of hydroxyl group-free polyfluorene (TPE-PF) and PDMS to obtain ternary composites. We found that the luminescence from this composite was not significantly affected by the silica content or external temperature. We synthesized temperature-responsive AIE materials without changing the concentration or aggregation state of the AIE molecules.

## 1. Introduction

It is well known that conjugated polymers exhibit interesting electronic and optical properties [1]. Conjugated polymers such as polythiophenes and polyfluorenes have attracted much attention as luminescent polymers mainly for organic light-emitting diodes (OLEDs) and biomarkers [2]. Many conjugated luminescent polymers show strong luminescence in a solution but are partially or completely quenched in an aggregated state. This effect is known as aggregation-caused quenching (ACQ) [3]. Such concentration quenching leads to a decrease in the luminous efficiency of the light-emitting device. In the case of biomarkers, aggregation with biomolecules and analytical agents reduces sensitivity. On the other hand, some molecules such as tetraphenylethene (TPE) emit light in a solid state but not in a solution. This phenomenon is called aggregation-induced emission (AIE) [4,5,6,7]. The widely accepted reason for the AIE effect is the restriction of intramolecular motion (RIM), which is composed of both the restriction of intramolecular rotation (RIR) and the restriction of intramolecular vibration (RIV). In a solution state, exited energy is consumed by molecular motion, and the release of energy from the excited state to the ground state is non-radiative deactivation. In an aggregated state, on the other hand, spatial constraints and interactions with surrounding molecules significantly suppress these molecular motions, inhibiting the non-radiative deactivation pathway and resulting in luminescence [8].

AIE polymers may be superior to lower-molecular-weight AIE molecules in terms of prosessability, further functionalization, and thermal stability. Polymeric AIE molecules have many useful applications as new optical materials and have been widely investigated [9,10,11]. A typical AIE polymer is one in which a segment with an AIE functional group is introduced into the main chain or side chain moiety of the polymer. In most cases, an AIE–functional group has a propeller-shaped structure with rotatable periphery phenyl rings such as TPE and hexaphenylsilole. And such a polymer can be prepared by ether the polymerization of the AIE–functional monomer or post-polymerization modification. The main chain of the polymer may be a conjugated structure or an unconjugated structure. Compared to unconjugated AIE polymers, conjugated AIE polymers are expected to have higher luminescence efficiency. For example, Wu et al. synthesized a new series of TPE-containing conjugated polyfluorene copolymers through a palladium-catalyzed Suzuki polycondensation reaction and found that all polymers exhibited AIE properties thanks to the TPE moieties [12].

We have reported the incorporation of emitting polymers into silica as new emitting materials [13,14,15]. For example, we introduced pendant hydroxyl groups into polyfluorene and carried out a sol-gel reaction of tetraethoxysilane (TEOS) in the presence of hydroxyl-functionalized polyfluorene to obtain an organic/inorganic hybrid retaining the optical properties of the embedded polyfluorene [16]. Since we observed polymer aggregation in the resulting silica when we carried out the sol-gel reaction of TEOS in the presence of a conjugated polymer without hydroxyl functionalities, hydrogen bonding between pendant hydroxyl groups and silanol groups in silica played an important role for homogeneous hybrid formation. Further, we successfully immobilized emitting polyfluorenes into silicone resin by the sol-gel reaction of TEOS in the presence of polyfluorene with pendant ethoxysilyl groups and silanol-terminated polydimethylsiloxane. Homogeneous hybridization proceeded because of the covalent bond formation between ethoxysilyl and silanol groups [17]. By immobilizing the luminescent polymer in a transparent solid matrix, energy transfer did not occur because, unlike in the solution state, the molecules lose their mobility and no intermolecular contact occurs. In other words, it corresponds to an isolated dispersion of molecules in a frozen state. On the other hand, if the solid matrix is soft like silicone, molecular motion is possible to some extent.

So far, the external environment affecting the aggregation-induced effect of polymers has been exclusively investigated in terms of solution concentration and solvent composition. We were interested in immobilizing AIE polyfluorene carrying TPE moieties into a solid matrix in an isolated dispersion. Our idea was to control the degree of the rotation of phenyl rings by changing the softness of the solid matrix. We hypothesized that by changing the softness of the external environment in which the aggregation-inducing molecules reside, the ease of intramolecular rotation can be changed and, as a result, the aggregation-inducing effect can be controlled. The solid matrix we focused on was a hybrid composed of PDMS and silica. Recently, hybrid materials composed of PDMS and SiO_2_ have attracted much attention as biomaterials [18,19,20], photonic materials [21], and coating materials [22,23,24]. This diversity of applications is related to the flexibility of the materials ranging from a hard solid to a rubber-like substance.

The chemical structures of the compounds used in this study are shown in Figure 1. We synthesized AIE polyfluorene (TPE-PF-OH) which carries both TPE as an AIE active moiety and pendant hydroxyl groups which are capable of interacting with silica. Our preliminary experiment showed that the sol-gel reaction of polydiethoxysiloxane (PDEOS) in the presence of TPE-PF-OH and PDMS gave a translucent solid, indicating that the aggregation of TPE-PF-OH molecules took place in the solid matrix. Therefore, we prepared PDMS containing pendant hydroxyl groups (PDMS-OH), which can interact with TPE-PF-OH through hydrogen bonding. The sol-gel reactions of PDEOS were carried out in the presence of TPE-PF-OH and PDMS-OH to obtain TPE-PF-OH/PDMS-OH/SiO_2_ hybrids with different amounts of silica. We measured the emission spectra of the resulting ternary TPE-PF-OH/PDMS-OH/SiO_2_ hybrids to examine the effect of silica content and temperature on the luminescence intensity from the hybrid. For comparison, the sol-gel reactions of PDEOS were carried out in the presence of hydroxyl-free polyfluorene and PDMS to obtain ternary composites, in which AIE polyfluorene molecules were embedded in the solid matrix in the aggregated state. The optical properties of TPE-PF-OH/PDMS-OH/SiO_2_ hybrids were compared with those of TPE-PF/PDMS/SiO_2_ composites.

## 2. Materials

### 2.1. Reagents

1,2-Bis(4-bromophenyl)-1,2-diphenylethene [25], 2,7-dibromo-9,9-bis(6-(2-tetrahydropyranyloxy)hexyl)fluorene [16], and 2,7-dibromo-9,9-dihexylfluorene [26] were prepared according to the reported procedures. The platinum–divinyltetramethyldisiloxane complex (3.0% Pt in vinyl-terminated PDMS) (SIP 6830), trimethylsilyl-terminated poly(dimethylsiloxane-co-methylhydrosiloxane) (PDMS-H) (molecular weight, 20,000–25,000; methylhydrosiloxane, 4–6 mol%), and trimethylsilyl-terminated polydimethylsiloxane (PDMS) (molecular weight, 26,000) were purchased from Gelest, Inc. (Morrisville, PA, USA) Tris(tris [3,5-bis(trifluoromethyl)phenyl]phosphine)palladium (0) was purchased from Wako Pure Chemical Industries, Ltd. (Osaka, Japan) Aliquat 336 was purchased from Aldrich (St. Louis, MO, USA). All other reagents were obtained from commercial sources and used as received.

### 2.2. Compounds

#### 2.2.1. 1,2-Diphenyl-1,2-bis(4-(4,4,5,5-tetramethyl-1,3,2-dioxaborolan-2-yl)phenyl)ethene (**1**)

To a mixture of 1,2-bis(4-bromophenyl)-1,2-diphenylethene (2.8 g, 5.7 mmol), bis(pinacolato)diboron (3.8 g, 15 mmol) and KOAc (4.4 g, 45 mmol), degassed dioxane (50 mL) and [1,1′-bis(diphenylphosphino)ferrocene]palladium(II) dichloride (Pd(dppf)Cl_2_, 60 mg) were added and the reaction mixture was stirred at 80 °C for 24 h. After cooling to room temperature, the reaction mixture was diluted with chloroform and washed with water. The organic layer was dried over anhydrous magnesium sulfate and placed under reduced pressure to remove the solvent. The residue was purified by column chromatography using a mixture of dichloromethane and hexane (2:1 *v*/*v*) as an eluent to give 2.2 g (67%) of compound **1** as a white solid; ^1^H NMR (500 MHz, CDCl_3_, δ): 7.53 (d, *J* = 7.2 Hz, 4H), 7.1–7.0 (m, 14H), 1.32 (s, 24H); ^13^C NMR (125 MHz, CDCl_3_, δ):146.9, 143.5, 141.4, 134.1, 131.4, 130.8, 127.8, 127.7, 126.6, 83.7, 24.9; IR (KBr, cm^−1^): 2977, 1607; Anal. calcd. for C_38_H_42_B_2_O_4_: C 78.10, H 7.24; found: C 78.19, H 7.16.

#### 2.2.2. TPE-PF-OTHP

A mixture of **1** (584 mg, 1.00 mmol), 2,7-dibromo-9,9-bis(6-(2-tetrahydropyranyloxy)hexyl)fluorene (**2**) (139 mg, 0.20 mmol), 2,7-dibromo-9,9-dihexylfluorene (**3**) (394 mg, 0.80 mmol), toluene (20 mL), 2 mol/L aqueous Na_2_CO_3_ (5 mL), and tris(tris [3,5-bis(trifluoromethyl)phenyl]phosphine)palladium (0) (10 mg) was deaerated by bubbling argon at least 10 min. The reaction mixture was heated at 90 ºC for 72 h under argon and then poured into methanol. The precipitated polymer was purified by washing for 2 days in a Soxhlet apparatus with acetone to remove oligomers and catalyst residues to obtain 570 mg (81%) of TPE-PF-OTHP as a yellow powder; UV-vis (in THF): λ_max_ = 356 nm; IR (KBr, cm^−1^): 2923, 2836, 1455, 693; GPC: *M*_w_ = 42,000, *M*_w_/*M*_n_ = 2.5.

#### 2.2.3. TPE-PF-OH

A mixture of TPE-PF-OTHP (0.29 g), 30 mL of THF, and 5 mL of 10% hydrochloric acid was stirred at 40 °C for 20 h. The reaction mixture was diluted with chloroform and washed with water. The organic layer was dried over anhydrous magnesium sulfate and placed under reduced pressure to remove the solvents. The residue was dissolved in a small amount of chloroform and then re-precipitated into methanol to obtain 0.18 g (63%) of TPE-PF-OH as a yellow powder; UV-vis (in THF): λ_max_ = 359 nm; IR (KBr, cm^−1^): 2929, 2845, 1460, 722; GPC: *M*_w_ = 60,300, *M*_w_/*M*_n_ = 1.9.

#### 2.2.4. PDMS-OH

To a mixture of PDMS-SiH (5.0 g, SiH = 3.5 mmol) and allyl alcohol (0.81 g, 14 mmol) in 20 mL of toluene was added platinum catalyst (SIP 6830, 10 mg), and the reaction mixture was heated at 70 °C for 24 h. The rection mixture was pored into methanol to precipitate 3.8 g (71%) of PDMS-OH as colorless viscous oil; IR (NaCl, cm^−1^): 2967, 1266, 1100, 1031, 797.

#### 2.2.5. TPE-PF

A mixture of **1** (0.45 g, 0.77 mmol), **3** (0.38 g, 0.77 mmol), THF (20 mL), 2 mol/L aqueous Na_2_CO_3_ (4 mL), and tris(tris [3,5-bis(trifluoromethyl)phenyl]phosphine)palladium (0) (10 mg) was deaerated by bubbling argon at least 10 min. The reaction mixture was heated at 90 ºC for 72 h under argon and then precipitated into methanol. The polymer was filtered and purified by washing for 2 days in a Soxhlet apparatus with acetone to remove oligomers and catalyst residues to obtain 0.40 g (78%) of TPE-PF as a yellow powder; UV-vis (in THF): λ_max_ = 369 nm; IR (KBr, cm^−1^): 2922, 2864, 1465, 703; GPC: *M*_w_ = 13,200, *M*_w_/*M*_n_ = 2.8.

#### 2.2.6. TPE-PF-OH/PDMS-OH/SiO_2_ Hybrid

In a typical example, to a mixture of PDMS-OH (0.60 g) and PDEOS (0.37 g), a solution of TPE-PF-OH (0.1 mg) in 10 mL of THF and 5 μL of dibutyltin dilaurate was added. The reaction mixture was stirred for 1 h and then allowed to stand at room temperature for one week to obtain 0.75 g of a pale-yellow transparent solid; IR (KBr, cm^−1^): 2966, 1252, 1068, 1025, 708.

#### 2.2.7. TPE-PF/PDMS/SiO_2_ Composite

In a typical example, to a mixture of PDMS (0.60 g) and PDEOS (0.37 g), a solution of TPE-PF (0.1 mg) in 10 mL of THF and 5 μL of dibutyltin dilaurate was added. The reaction mixture was stirred for 1 h and then allowed to stand at room temperature for one week to obtain 0.75 g of a pale-yellow translucent solid; IR (KBr, cm^−1^): 2967, 1251, 1068, 1021, 711.

### 2.3. Measurements

Nuclear magnetic resonance spectra (NMR) were recorded on 500 MHz for ^1^H spectra and 125 MHz for ^13^C spectra (ECZ500R, JEOL, Tokyo, Japan). The analysis was conducted at room temperature. The samples were dissolved in CDCl_3_, with tetramethylsilane (TMS) serving as the internal standard. Photoluminescence spectra were recorded on a HAMAMATSU Multi Channel Analyzer PMA-11 (Hamamatsu, Japan). The measurement was conducted at the exciting wavelength of 365 nm. Fourier transform infrared (FTIR) spectra and UV-vis spectra were recorded on JASCO FT/IR-4100 (Tokyo, Japan) and SHIMADZU UV-2550 (Kyoto, Japan), respectively. Elemental analysis was carried out using YANACO CHN-corder MT-5 (Kyoto, Japan). Gel permeation chromatography (GPC) was carried out on a Tosoh HLC-8020 chromatograph equipped with polystyrene gel columns (Tosoh Multipore HXL-M, Tokyo, Japan; exclusion limit = 2 × 10^6^, 300 × 7.8 mm) and refractive/ultraviolet dual mode detectors. Tetrahydrofuran (THF) was used as the eluent at a flow rate of 1.0 mL/min. The calibration curves for GPC analysis were obtained using polystyrene standards.

## 3. Results and Discussion

### 3.1. Preparation of TPE-PF-OH

The synthetic pathway for PTE-PF-OH is shown in Figure 2. The key compound is a fluorene derivative **2** in which hydroxyl groups are protected by tetrahydropyranyl (THP) groups because the THP group, like other acetals and ketals, is inert under basic conditions during the Suzuki coupling reaction. The ternary copolymerization of **1**, **2** and **3** was carried out in the presence of a palladium catalyst. Since we already found that the introduction of 20 mol% hydroxyl group-containing monomer allowed for homogeneous mixing with silica [16], we added 20 mol% of monomer **2**. The ^1^H NMR spectrum of the resulting TPE-PF-OTHP is shown in Figure 3. The peak at 0.6–0.8 ppm is due to the CH_3_ protons of the hexyl group. The peaks at 4.5 and 3.8–3.2 ppm are assigned as CH and OCH_2_ protons, respectively. The polymer composition of TPE-PF-OTHP was determined by ^1^H NMR through the peak area ratio between the signals coming from CH_3_ protons and those belonging to OCH_2_ protons to be m:n = 19:81 which corresponded well with the expected value on the basis of the monomer feed ratio (m:n = 1:4). The THP group was then removed by conventional acid treatment in THF. Figure 4 shows ^1^H NMR of TPE-PF-OH. The peaks due to THP groups disappeared completely, indicating the complete conversion of TPE-PF-OTHP to TPE-PF-OH.

Figure 5 shows the emission spectra of TPE-PF-OH in a diluted THF solution (0.1 mg/mL) and from thin film. Weak emission was observed in the THF solution (Figure 5a). This is probably because the intramolecular rotation process of the four phenyl rings in the TPE moieties in conjugated polymer may be limited in some degree, and weak emission still present in the solution state. On the other hand, much stronger emission was observed from the thin film of TPE-PF-OH (Figure 5b), indicating that TPE-PF-OH exhibits a typical AIE property [3].

### 3.2. Preparation of PDMS-OH

In order to immobilize TPE-PF-OH in a PDMS/SiO_2_ matrix in an isolated and dispersed state, we introduced pendant hydroxyl groups onto PDMS. The introduced hydroxyl groups on PDMS should help homogeneous mixing with emissive polymer with hydroxyl functionalities. We synthesized poly[dimethylsiloxane-*co*-methyl(3-hydroxypropyl)siloxane] (PDMS-OH) via the hydrosilylation reaction of trimethylsilyl-terminated poly(dimethylsiloxane-*co*-methylhydrosiloxane)] (PDMS-H) with an allyl alcohol in the presence of a platinum catalyst. Figure 6 shows the ^1^H NMR spectra of PDMS-OH with that of the starting PDMS-H. PDMS-H shows an absorption peak at 4.6 ppm due to Si-H groups. After the hydrosilylation reaction, the peak at 4.6 ppm disappeared completely, while new peaks emerged at 3.3, 1.6, and 0.3 ppm coming from SiCH_2_CH_2_CH_2_OH groups, indicating that Si-H groups were successfully converted to Si-CH_2_CH_2_CH_2_OH groups. Figure 7 shows the IR spectra of PDMS before and after the hydrosilylation reaction. After hydrosilylation, the peak at 2155 cm^−1^ due to Si-H groups disappeared completely.

### 3.3. Preparation of TPE-PF-OH/PDMS-OH/SiO_2_ and TPE/PDMS/SiO_2_

We synthesized three TPE-PF-OH/PDMS-OH/SiO_2_ hybrids with different silica contents. Although the preparation of hybrid elastomers composed of PDMS and SiO_2_ was reported by the sol-gel reaction of tetraethoxysilane (TEOS) in the presence of hydroxy-terminated poly(dimethylsiloxane) [27], our preliminary experiments revealed that it was difficult to control the silica content in the hybrid due to volatilization of TEOS during the sol-gel reaction. We utilized poly(diethoxysilane) (PDEOS) as a silica precursor for a nonaqueous sol-gel reaction [28]. PDEOS is known as a non-volatile oligomeric form of TEOS. The sol-gel reaction conditions are summarized in Table 1. Since 1 g of PDEOS changes to about 0.41 g of silica after elimination of ethanol, silica content (wt%) in the hybrid can be easily calculated. The ternary hybrids obtained were transparent solid without polymer aggregation. The hybrid with 60 wt% silica was a hard solid, while the hybrid with 20 wt% silica was a rather rubbery solid.

For the control experiment, we synthesized AIE polyfluorene without hydroxyl functionalities by the Suzuki polycondesation reaction between **1** and **3** to obtain TPE-PF which is an alternating copolymer of TPE and 9,9-dihexylfluorene. The ^1^H NMR of TPE-PF is shown in Figure 8. Then, we carried out the sol-gel reaction of PDEOS in the presence of TPE-PF and PDMS to obtain TPE-PF/PDMS/SiO_2_ composites with different SiO_2_ content. Table 2 summarizes the sol-gel reaction conditions. Since there is no interaction such as hydrogen bonding, electrostatic interaction, or π–π interaction among TPE-PF, PDMS, and silica, the resulting solid was an opaque solid, indicating that TPE-PF molecules aggregate in the solid matrix. This indicates that phenyl rings at a TPE moiety cannot rotate any more.

### 3.4. Effect of Silica Content on Emission

Figure 9a shows the emission spectra of TPE-PF-OF/PDMS-OH/SiO_2_ hybrids with different silica contents at room temperature. It was obvious that emission intensity increased with the increase in silica content. This is reasonably explained by considering the restriction of intramolecular rotation of phenyl rings of TPE groups. It becomes difficult for phenyl rings to rotate when the solid matrix becomes harder [27].

Figure 9b shows the emission spectra of TPE-PF/PDMS/SiO_2_ composites with different silica contents at room temperature. Although the emission intensity decreased with the decrease in silica content, the effect was less remarkable when compared with that for the TPE-PF-OH/PDMS-OH/SiO_2_ system. This is because phenyl rings cannot rotate even if the matrix becomes soft.

### 3.5. Effect of Temperature on Emission

Next, we examined the effect of temperature on emission property. Figure 10a shows the emission spectra of the TPE-PF-OH/PDMS-OH/SiO_2_ hybrid with 40 wt% silica at various temperatures. The fluorescence intensity decreased with the increase in temperature. The observed decrease in emission intensity can be explained by considering an easier rotation of pheny rings at a higher temperature.

Figure 10b shows the emission spectra of TPE-PF/PDMS/SiO_2_ composites at various temperatures. Similar to the TPE-PF-OH/PDMS-OH/SiO_2_ system, a decrease in emission intensity was observed as the external temperature increased. However, the decrease in emission intensity with increasing temperature is less pronounced than the TPE-PF-OH/PDMS-OH/SiO_2_ system.

These experimental results show that the fluorescence intensity of TPE-PF/PDMS/SiO_2_ composites is not significantly affected by the silica content or temperature. This is because TPE-PF molecules exist in an aggregated state due to the lack of interaction with the solid matrix, making it difficult for phenyl rings to rotate, supporting that the rotation of phenyl rings plays an important role in bringing about the AIE effect. There are not many reported papers of thermo-responsive AIE molecules. Liu et al. synthesized polyurethanes with soft and hard segments using AIE-active tetra-aniline derivative as the hard segment, and they found that these polyurethanes exhibited temperature-dependent fluorescent characteristics [29]. Ma et al. synthesized poly(N-isopropylacrylamide) (PNIPAAm) with an AIE moiety to observe thermo-induced emission due to the aggregation of PNIPAAm chains [30]. Our ternary hybrid containing AIE polyfluorene is another example of a thermo-responsive AIE material. The mechanism of the thermo-responsibility is coming from the change of softness of the transparent solid matrix in which AIE molecules are imbedded.

## 4. Conclusions

We synthesized two AIE polyfluorenes with a TPE moiety. One (TPE-PF-OH) had pendant hydroxyl groups while the other (TPE-PF) did not. And we converted PDMS-H to PDMS-OH with hydroxypropyl groups. We carried out the sol-gel reactions of PDEOS in the presence of TPE-PF-OH/PDMS-OH or TPE-PF/PDMS. The former gave transparent solids while the latter gave translucent solids. We examined emission properties by changing the silica content and temperature for these ternary systems. The effects of the silica content and temperature on fluorescence intensity differed significantly between the TPE-PF-OH/PDMS-OH/SiO_2_ and TPE-PF/PDMS/SiO_2_ systems. The luminescence intensity from TPE-PF-OH/PDMS-OH/SiO_2_ was greatly affected by the silica content and temperature. On the other hand, the luminescence from TPE-PF/PDMS/SiO_2_ was not significantly affected by the silica content or temperature. These results can be reasonably explained by considering the intramolecular motion of phenyl rings at a TPE moiety. We demonstrated that the luminescence properties of an AIE polymer can be altered by isolating and dispersing it in a transparent solid matrix of which the flexibility can be changed by external stimuli. Such materials are expected to be new temperature-responsive optical materials based on AIE phenomena.

## Figures and Tables

**Figure 1 polymers-16-03331-f001:**
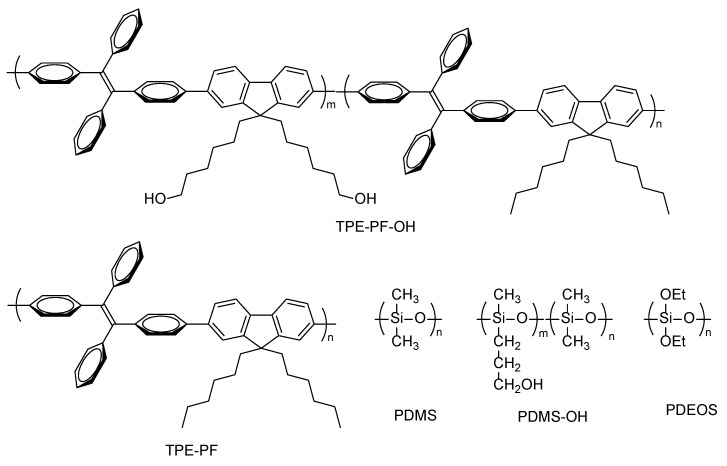
Chemical structure of the compounds used in this study.

**Figure 2 polymers-16-03331-f002:**
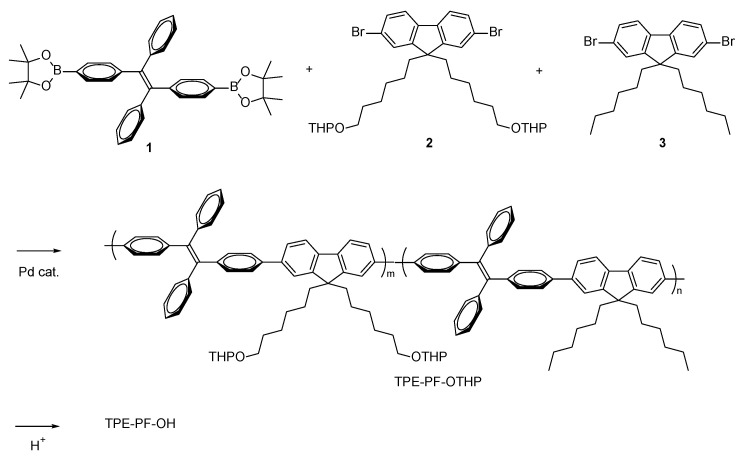
Synthetic pathway for TPE-PF-OH.

**Figure 3 polymers-16-03331-f003:**
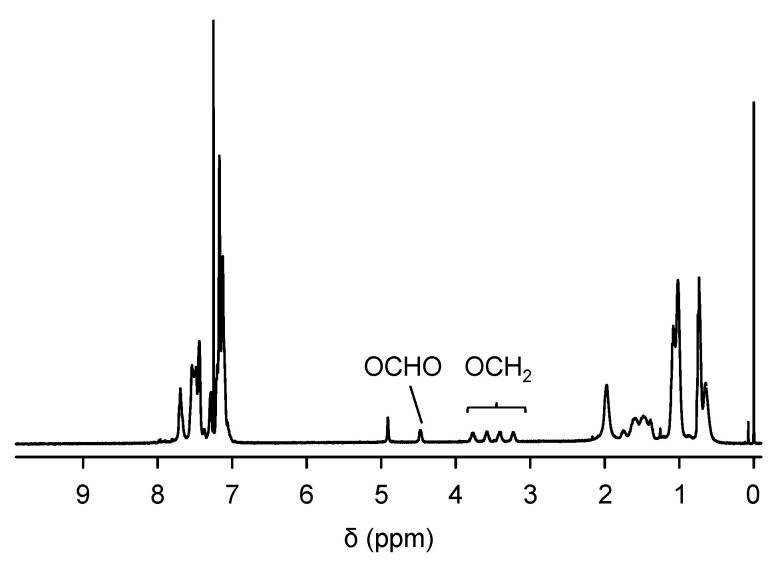
^1^H NMR spectrum of TPE-PF-OTHP.

**Figure 4 polymers-16-03331-f004:**
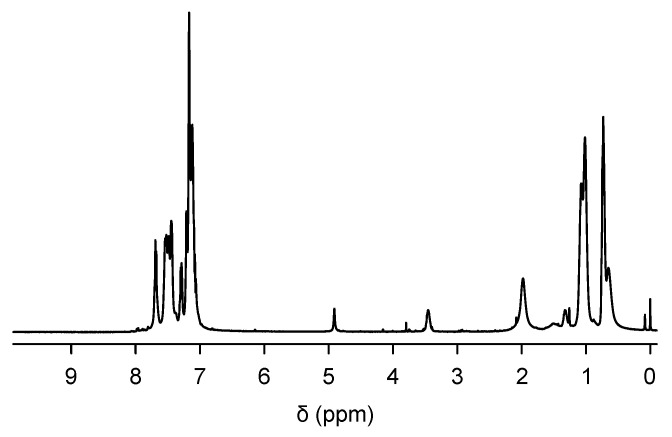
^1^H NMR spectrum of TPE-PF-OH.

**Figure 5 polymers-16-03331-f005:**
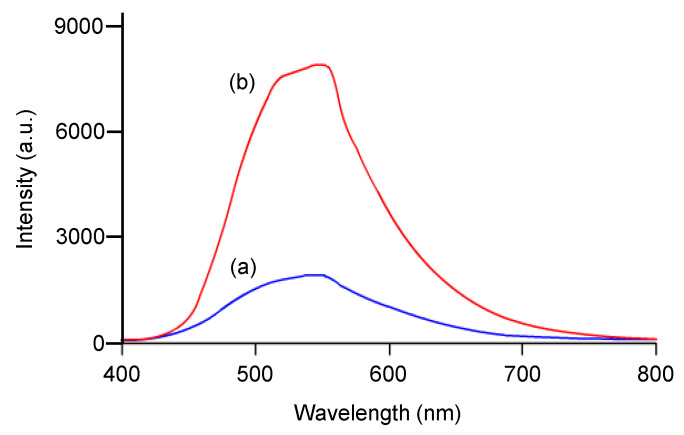
Emission spectra of TPE-PF-OH (a) in a THF solution (0.1 mg/mL) and (b) from thin film.

**Figure 6 polymers-16-03331-f006:**
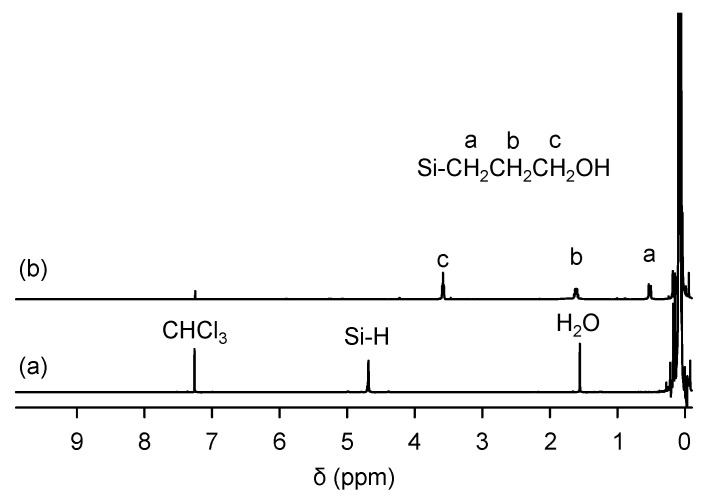
^1^H NMR spectra of (a) PDMS-H and (b) PDMS-OH.

**Figure 7 polymers-16-03331-f007:**
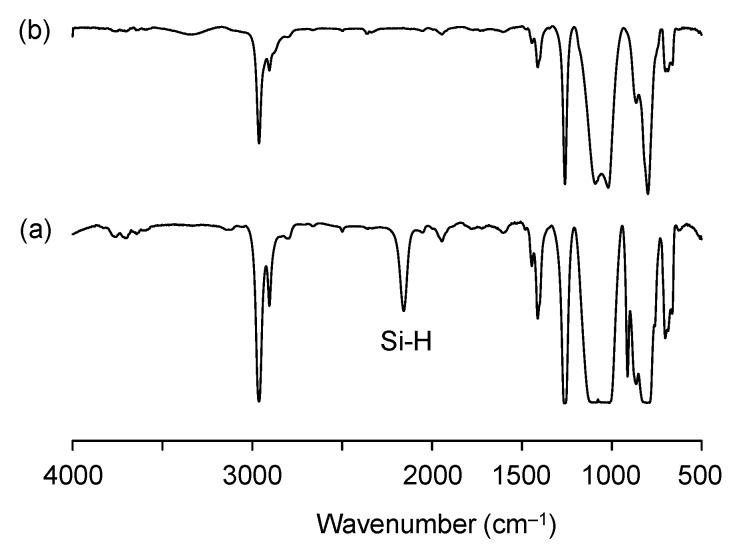
IR spectra of (a) PDMS-H and (b) PDMS-OH.

**Figure 8 polymers-16-03331-f008:**
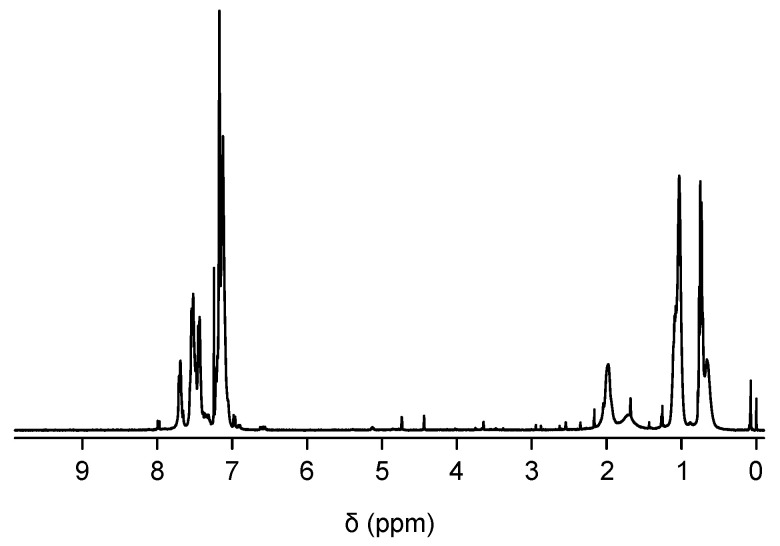
^1^H NMR spectrum of TPE-PF.

**Figure 9 polymers-16-03331-f009:**
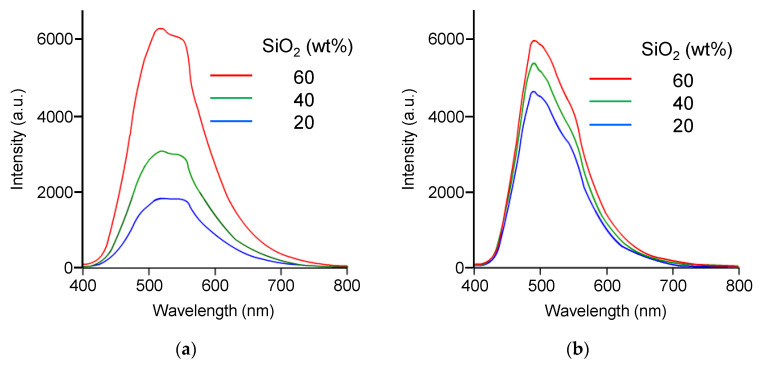
Emission spectra from (**a**) the TPE-PF-OH/PDMS-OH/silica hybrid and (**b**) TPE-PF/PDMS/silica composite at various silica contents.

**Figure 10 polymers-16-03331-f010:**
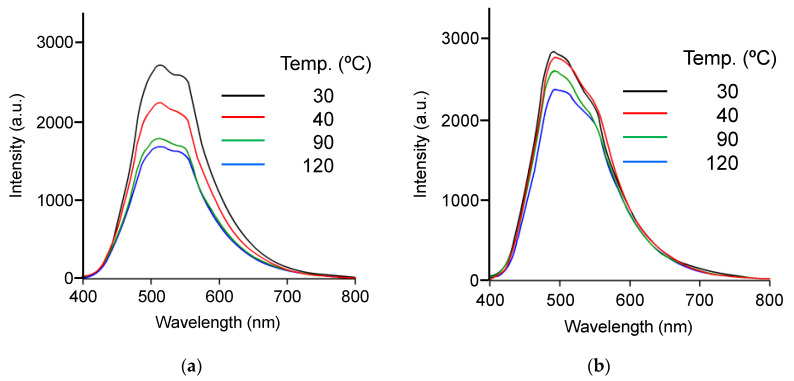
Emission spectra from (**a**) the TPE-PF-OH/PDMS-OH/SiO_2_ hybrid and (**b**) TPE-PF/PDMS/SiO_2_ composite at various temperatures.

**Table 1 polymers-16-03331-t001:** So-gel reaction conditions ^1^ for PTE-PF-OH/PDMS-OH/SiO_2_ hybrids.

Entry	PDMS-OH, g	PDEOS, g	SiO_2_ Content, wt%
1	0.60	0.37	20
2	0.45	0.73	40
3	0.30	1.1	60

^1^ TPE-PF-OH = 0.1 mg, (C_4_H_9_)_2_Sn(OCOC_11_H_23_)_2_ = 5 μL, THF = 10 mL, temp., = rt, time = 1 week.

**Table 2 polymers-16-03331-t002:** So-gel reaction conditions ^1^ for PTE-PF/PDMS/SiO_2_ composites.

Entry	PDMS, g	PDEOS, g	SiO_2_ Content, wt%
1	0.60	0.37	20
2	0.45	0.73	40
3	0.30	1.1	60

^1^ TPE-PF = 0.1 mg, (C_4_H_9_)_2_Sn(OCOC_11_H_23_)_2_ = 5 μL, THF = 10 mL, temp., = rt, time = 1 week.

## Data Availability

The original contributions presented in this study are included in the article. Further inquiries can be directed to the corresponding author.

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
