# Peer review of "The Synthesis and Optical Property of a Ternary Hybrid Composed of Aggregation-Induced Luminescent Polyfluorene, Polydimethylsiloxane, and Silica"

_polymers, 2024, doi:10.3390/polym16233331_

Round 1
Reviewer 1 Report
Comments and Suggestions for Authors
The manuscript "Synthesis and Optical Property of Ternary Hybrid Composed of Aggregation-Induced Luminescent Polyfluorene, PDMS, and Silica" reports preparation of hybrid materials based on polydiethoxysiloxane (PDEOS), hydroxyl-functionalized AIE polyfluorene (TPE-PF-OH), polydimethylsiloxane with hydroxyl groups (PDMS-OH) and SiO2 to explore the aggregation-induced emission function of them.
Abstract better emphasizes the motivation of this study. It is not clear the dependence of luminescence on SiO2 particles, which was previously reported in references 13-15.
Keywords: delete: "Restriction of intramolecular rotation"
Introduction: There are a lot of studies reported by authors. Literature also must cited, the advantages of the materials developed by authors over other studies, the limits in addressing such hybrids, etc.
Section 2.2. IR data for hybrids described at 2.2.6 and 2.2.7 are missing.
Section 3. Fig. 5 - Y axis is missing!
line 216-add a reference to support the comment.
line 231. The complete hydrosilylation also must be demonstrated by IR spectrum!
line 237: the content of silica must be mentioned, not only as percent. Catalyst was not used? (section 3.3.)
section 3.4. a reference must be added at line 257.
Fig. 7. Y axis is missing!
The authors also stated at lines 85-87 that hydrogen bonds between PDMS-OH and TPE-PF-OH also occur, but this statement is not supported by experimental data.
Section 3.5. Y axis is missing in Fig. 8. How was appreciated the fluorescence modification since the values are missing!!!
Some literature comparisons must be added!
Section 3.6. line 277- what interactions would have been expected?
In my opinion, this paper includes a series of experimental results, but these are not sufficient discussed and proved, mostly by assumption. Based on these comments I suggest this paper to be major revised before the acceptance!
Author Response
1. Summary |
|
|
Thank you very much for taking your time to review this manuscript. Please find the detailed responses below and the corresponding revisions in the re-submitted file. |
||
|
||
3. Point-by-point response to Comments and Suggestions for Authors |
||
Comment 1: Abstract better emphasizes the motivation of this study. It is not clear the dependence of luminescence on SiO2 particles, which was previously reported in references 13-15. |
||
Response 1: Thank you for your comment. We added our new finding in abstract (line 20-). |
||
|
||
Comment 2: Keywords: delete: "Restriction of intramolecular rotation" |
||
Response 2: We deleted " Restriction of intramolecular rotation " in Keywords. (line 22) |
||
|
||
Comment 3: Introduction: There are a lot of studies reported by authors. Literature also must cited, the advantages of the materials developed by authors over other studies, the limits in addressing such hybrids, etc. |
||
Response 3: Thank you for your comment. We revised Introduction. We described the characteristics of our materials reported previously. (line 69-) |
||
|
||
Comment 4: Section 2.2. IR data for hybrids described at 2.2.6 and 2.2.7 are missing. |
||
Response 4: Thank you for pointing it out. We added IR data for hybrids. (line 170-, line 176-) |
||
|
||
Comment 5: Section 3. Fig. 5 - Y axis is missing! |
||
Response 5: We added Y axis. |
||
|
||
Comment 6: line 216-add a reference to support the comment. |
||
Response 6: We added reference. (line 223) |
||
|
||
Comment 7: line 231. The complete hydrosilylation also must be demonstrated by IR spectrum! |
||
Response 7: Thank you for pointing it out. We added IR spectra before and after hydrosilylation reaction to make it easier for readers to understand. (line 238-) |
||
|
||
Comment 8: line 237: the content of silica must be mentioned, not only as percent. Catalyst was not used? (section 3.3.) |
||
Response 8: Thank you for your comment. Since the amount of catalyst is very low, we ignored the amount of catalyst when determining silica content. The TEM observation of the hybrids and composites are now undergoing. |
||
|
||
|
||
Comment 9: section 3.4. a reference must be added at line 257. |
||
Response 9: We added a reference. (line 281) |
||
|
||
Comment 10: Fig. 7. Y axis is missing! |
||
Response 10: We added Y axis. |
||
|
||
Comment 11: The authors also stated at lines 85-87 that hydrogen bonds between PDMS-OH and TPE-PF-OH also occur, but this statement is not supported by experimental data. |
||
Response 11: Thank you for the comment. We checked IR spectrum of the hybrid to get an evidence of hydrogen bonding. However, it was difficult probably because of the small amount of TPE-PF-OH in the hybrid. Instead, we added result of our preliminary experiment using PDMS without hydroxyl group. (line 88-) |
||
|
||
Comment 12: Section 3.5. Y axis is missing in Fig. 8. How was appreciated the fluorescence modification since the values are missing!!! |
||
Response 12: We added Y axis. |
||
|
||
Comment 13: Some literature comparisons must be added! |
||
Response 13: Thank you for the suggestion. We added examples of thermo-responsive AIE materials and mentioned the difference between our study and these published works. (line 302-) |
||
|
||
Comment 14: Section 3.6. line 277- what interactions would have been expected? |
||
Response 14: We specified expected interactions. (line 264) |

Reviewer 2 Report
Comments and Suggestions for Authors
In this work, the author synthesized two AIE polyfluorenes with TPE moiety. And they examined emission properties by changing silica content and temperature for these ternary systems.
Here are my concerns regarding this work:
1, In Figure 2, the name of TPE-PF-OTHP was labelled under the chemical structure. However, the structure has groups of OH, which is a mistake.
2, In the line of 253, the author mentioned "Figure 7a shows emission spectra of TPE-PF-OF/PDMS-OH/SiO2". However, here should be TPE-PF-OH/PDMS-OH/SiO2.
3 The contents of Figure 7b and Figure 8b had better be discussed with Figure 7a and Figure 8a. When the author discussed the Figure 7a, Figure 7b should be mentioned at the same time. Similar to Figure 8b.
4. What is the highlight of this work? In my opinion, there is no discussion regarding material application or novel design, findings, mechanism. The author had better think about the highlight of this work and more discussion is needed, instead of presenting results.
Author Response
1. Summary |
|
|
Thank you very much for taking your time to review this manuscript. Please find the detailed responses below and the corresponding revisions in the re-submitted file. |
||
|
||
3. Point-by-point response to Comments and Suggestions for Authors |
||
Comment 1: In Figure 2, the name of TPE-PF-OTHP was labelled under the chemical structure. However, the structure has groups of OH, which is a mistake. |
||
Response 1: Thank you for pointing it out. We amended the structure in Figure 2. |
||
|
||
Comment 2: In the line of 253, the author mentioned "Figure 7a shows emission spectra of TPE-PF-OF/PDMS-OH/SiO2". However, here should be TPE-PF-OH/PDMS-OH/SiO2. |
||
Response 2: Thank you for pointing it out. We changed TPE-PF-OF/PDMS-OH/SiO2 to TPE-PF-OH/PDMS-OH/SiO2. |
||
|
||
Comment 3: The contents of Figure 7b and Figure 8b had better be discussed with Figure 7a and Figure 8a. When the author discussed the Figure 7a, Figure 7b should be mentioned at the same time. Similar to Figure 8b. |
||
Response 3: Thank you for your valuable suggestion. We revised the contents of these Figures according to your suggestion. We discussed these two Figures at the same time. (line 282-, line 297-) |
||
|
||
Comment 4: What is the highlight of this work? In my opinion, there is no discussion regarding material application or novel design, findings, mechanism. The author had better think about the highlight of this work and more discussion is needed, instead of presenting results. |
||
Response 4: Thank you for your valuable comment. Our most important finding is to prepare a new thermo-responsible material based on AIE polymer. We mentioned it in abstract (line 20-), section 3.5 (line 302-), and conclusions (line 332-) to clarify our work. |

Round 2
Reviewer 1 Report
Comments and Suggestions for Authors
The authors provided sufficient data to support their work. The paper was improved according to the suggestions and it is acceptable in this form.
Reviewer 2 Report
Comments and Suggestions for Authors
The author answered all my questions and this work can be published now.